# ConGra: Benchmarking Automatic Conflict Resolution

## Abstract

Resolving conflicts from merging different software versions is a challenging task. To reduce the overhead of manual merging, researchers develop various program analysis-based tools which only solve specific types of conflicts and have a limited scope of application. With the development of language models, researchers treat conflict code as text, which theoretically allows for addressing almost all types of conflicts. However, the absence of effective conflict difficulty grading methods hinders a comprehensive evaluation of large language models (LLMs), making it difficult to gain a deeper understanding of their limitations. Furthermore, there is a notable lack of large-scale open benchmarks for evaluating the performance of LLMs in automatic conflict resolution. To address these issues, we introduce ConGra, a **CON**flict-**GRA**ded benchmarking scheme designed to evaluate the performance of software merging tools under varying complexity conflict scenarios. We propose a novel approach to classify conflicts based on code operations and use it to build a large-scale evaluation dataset based on 44,948 conflicts from 34 real-world projects. We evaluate state-of-the-art LLMs on conflict resolution tasks using this dataset. By employing the dataset, we assess the performance of multiple state-of-the-art LLMs and code LLMs, ultimately uncovering two counterintuitive yet insightful phenomena. ConGra will be released at https://github.com/xxx/ConGra.

## 1 Introduction

Code merging has become a challenging task for developers during project development and maintenance. Git, the most popular version control system (Spinellis (2012)), uses a text-based code-merging mechanism. Despite its efficiency, developers often struggle with manually merging different versions of code when Git fails to resolve conflicts automatically. These conflicts come from text or even the syntax and the functionality of the code from different versions.

To address conflict resolution, researchers leverage program analysis to achieve syntax-error-free code merging (Zhu et al. (2022); Sousa et al. (2018); Larsén et al. (2022); Shen et al. (2019); Apel et al. (2012; 2011)). Based on abstract syntax trees (AST), these tools merge AST vertices and edges to ensure syntax correctness and better merge results. Nevertheless, a conflict is still generated and need to be resolved by developers if the merge of AST nodes fails. With the development of language models and even LLMs, they are now applied to conflict resolution (Dinella et al. (2022); Zhang et al. (2022); Svyatkovskiy et al. (2022); Dong et al. (2023)). Trained on extensive datasets of prior and manually merged codes, these models predict suitable resolution for each conflicting code segment without manual efforts.

However, evaluating the performance of LLMs on conflict resolution tasks is challenging due to the wide variation in conflict difficulty and the lack of effective grading methods to reflect these differences. For example, ConflictBench (Shen & Meng (2024)) classifies conflicts based on the source of conflict-resolved code, such as from either merging candidate versions or newly introduced by developers. This classification does not accurately reflect the complexity of the conflicts. Additionally, the community lacks comprehensive conflict resolution benchmarks (discussed in Section 2.4), especially for extreme cases involving long code contexts.

To this end, we introduce ConGra, which is designed to evaluate code merging tools across a diverse range of merging scenarios and assess their ability to resolve conflicts of varying complexities.

We propose a novel approach to construct graded conflict dataset according to conflict's resolving complexity. We evaluated six state-of-the-art LLMs (three general LLMs and three code LLMs) on CONGRA to assess their abilities to resolve conflict in various merging scenarios. The dataset is constructed using 44,948 conflict cases sourced from 34 large-scale open-source projects written in C, C++, Java, and Python. The results show that LLMs with longer contexts support do not always yield better results compared to models with shorter contexts. Additionally, general LLMs (e.g. LLama3-8B and DeepSeek-V2) outperform specialized code LLMs in automatically resolving conflict. Besides, we will release our datasets and a benchmark at https://github.com/xxx/ConGra.

In summary, we made the following contributions:

1. We introduce the first classification approach to generate a complexity-graded conflict dataset. Using this approach, collected conflicts can be classified into seven categories.

2. We release a large-scale graded dataset for conflict resolution benchmarking. which contains 44,948 conflict cases from popular projects written in C, C++, Java, and Python.

3. We conduct the first comprehensive evaluation of LLM's performance on conflict resolution task, and find two thought-provoking counter-intuitive phenomena.

## 2 BACKGROUND

### 2.1 TASK DEFINITION OF AUTOMATIC MERGE CONFLICT RESOLUTION

Git offers multiple strategies to merge code (Atlassian (2022)). The most common and widely-used strategy is the three-way merging strategy (Vinkler (2023)). The fundamental concept behind three-way merging is to locate the most recent common ancestor ($O$) through the historical commit graph, given two merging candidates ($A$ and $B$), and to generate the automerged code version ($M$) based on the difference $A - O$ and the difference $B - O$. Before generating $M$, Git identifies multiple difference matching triplets $A_i - O_i - B_i$ by comparing $A - O$ and $B - O$. For each difference matching triplet, 1) if $A_i - O_i$ equals $B_i - O_i$, it indicates that both $A$ and $B$ have made identical changes to the same code segment. In this case, the difference block will be applied to $M$; 2) if $A_i - O_i$ is empty, but $B_i - O_i$ exists, it suggests that only $B$ has modified this code segment. Consequently, the difference block from $B$ will be applied to $M$, and vice versa; 3) if $A_i - O_i$ and $B_i - O_i$ are not empty nor equal, it indicates that $A$ and $B$ have made different modifications to the same code segment, resulting in a *conflict* in $M$. In this paper, we assume that $M$ always contains at least one conflict.

Due to the presence of conflicts, Git-based automated code merging may encounter exceptions, prompting developers to manually address them. Following conflict resolution, developers will release the resolved code version ($R$). In large projects, merging two versions results in numerous conflicts, involving substantial code modifications. This significantly raises developers' project maintenance costs (Vale et al. (2021)). To this end, various automatic conflict resolution (ACR) systems have been proposed to mitigate these challenges.

### 2.2 PROGRAM ANALYSIS-BASED ACR

Program analysis-based ACR (Zhu et al. (2022); Larsén et al. (2022); Sousa et al. (2018); Apel et al. (2011); Shen et al. (2019); Apel et al. (2012)) ensures the syntactic integrity of the merged code by merging on AST. As they implement the three-way merging strategy, conflicts inevitably arise for unmergeable AST nodes, requiring manual intervention.

To assess the performance of program analysis-based ACR, researchers focus on metrics like the number of generated conflicts, the accuracy of resolutions, and resource consumption. However, these tools often lack convincing evaluation datasets and performance comparisons. As shown in Table 1, tools like Spork, SafeMerge, and Mastery (Larsén et al. (2022); Sousa et al. (2018); Zhu et al. (2022)) evaluate all merging scenarios without classification, obscuring performance differences across conflict complexities. JDIME and FSTMerge (Apel et al. (2012; 2011)) separately categorize structural and textual merges, but this is essentially an ablation study, leaving the dataset's resolving complexity intertwined. IntelliMerge (Shen et al. (2019)) classifies merge scenarios but fails to differentiate these categories in the final evaluation, preventing performance comparisons

Table 1: Evaluation features implemented in 1) program analysis-based ACR (tagged with †); 2) machine learning-based ACR (tagged with ‡); 3) conflict resolution benchmark (tagged with ⌖). The symbol "-" means the feature is not applicable to the corresponding work's evaluation.

| Evaluation Features | $Feature_1$ | $Feature_2$ | $Feature_3$ | $Feature_4$ | $Feature_5$ | $Feature_6$ |
|---|---|---|---|---|---|---|
| IntelliMerge† | ✓ | ✗ | ✓ | ✓ | ✓ | ✗ |
| Spork† | ✗ | ✗ | ✓ | ✓ | ✓ | ✗ |
| SafeMerge† | ✗ | ✗ | ✓ | ✗ | ✗ | ✗ |
| FSTMerge† | ✓ | ✗ | ✓ | ✗ | ✗ | ✗ |
| Mastery† | ✗ | ✗ | ✓ | ✗ | ✓ | ✗ |
| JDIME† | ✓ | ✗ | ✓ | ✗ | ✗ | ✗ |
| MergeBERT‡ | ✗ | ✗ | - | ✓ | ✓ | ✗ |
| GMerge‡ | ✗ | ✗ | - | ✓ | ✓ | ✗ |
| DeepMerge‡ | ✗ | ✗ | - | ✓ | ✓ | ✗ |
| MergeGen‡ | ✗ | ✗ | - | ✓ | ✓ | ✗ |
| RPredictor‡ | ✗ | ✗ | ✓ | ✓ | ✓ | ✗ |
| ChatMerge‡ | ✗ | ✗ | ✓ | ✓ | ✓ | ✗ |
| MESTRE‡ | ✗ | ✗ | - | ✓ | ✓ | ✗ |
| ConflictBench⌖ | ✓ | ✗ | ✓ | ✓ | ✓ | ✗ |
| CONGRA ⌖ | ✓ | ✓ | ✓ | ✓ | ✓ | ✓ |

$Feature_1$: Classify dataset; $Feature_2$: Assess performance under different graded conflicts; $Feature_3$: Record number of generated conflicts; $Feature_4$: Calculate precision of the generated conflict resolution; $Feature_5$: Calculate accuracy of the generated conflict resolution; $Feature_6$: Assess the whole code similarity instead of using exact string matching.

across scenarios. Additionally, these tools evaluate the quality of generated resolutions by exact string matching with the manual resolution (ground truth). However, this approach is inadequate for measuring code similarity, as it is influenced by programming style and code structure.

## 2.3 MACHINE LEARNING-BASED ACR

Machine learning-based ACR (Zhang et al. (2022); Dong et al. (2023); Svyatkovskiy et al. (2022); Dinella et al. (2022); Aldndni et al. (2023); Shen et al. (2023); Elias et al. (2023)) leverages a vast number of code merge examples during pre-training and harnesses the language-aware capabilities of machine learning models to understand the intricacies of code merging. Once trained, these models offer resolution suggestions for merge conflicts. They does not rely on the three-way merge strategy, eliminating majority of the need for manual conflict resolution.

Table 1 illustrates the features implemented in machine learning-based ACRs' evaluation. Machine learning-based ACRs must provide accurate resolutions for all conflicts, underscoring the importance of resolution correctness. Prior research efforts have primarily concentrated on improving two key metrics: precision and accuracy. Accuracy measures the percentage of total conflicts for which these tools produce the correct resolution, while precision indicates the percentage of correct resolutions among all resolution suggestions provided by the tools. By the same token as depicted in Section 2.2, the matching algorithm between generated resolution and ground truth are exact string matching, which cannot reflect the real code similarity. Furthermore, existing language model-based ACR test datasets also lack classification of merge scenarios based on different complexities, hindering the assessment of model performance across various levels of code merging tasks' difficulty.

## 2.4 CONFLICT RESOLUTION BENCHMARKS

ConflictBench (Shen & Meng (2024)) is the only benchmark for software code merging evaluation to date. It categorizes merging scenarios by resolution types and uses three metrics with an exact string matching algorithm to evaluate merging tools. However, as shown in Table 1, its classification strategy doesn't fully capture the complexity of merging scenarios. For example, conflicts can be resolved by retaining all edits from $A$ or $B$, or by introducing new edits. ConflictBench categorizes these into three groups, but the complexity boundaries between them are often blurred. Additionally,

ConflictBench's datasets include only 180 merging cases, and further classification limits the data volume in each category, potentially leading to inaccurate evaluation results.

# 3 BENCHMARKING PIPELINE

## 3.1 OVERVIEW

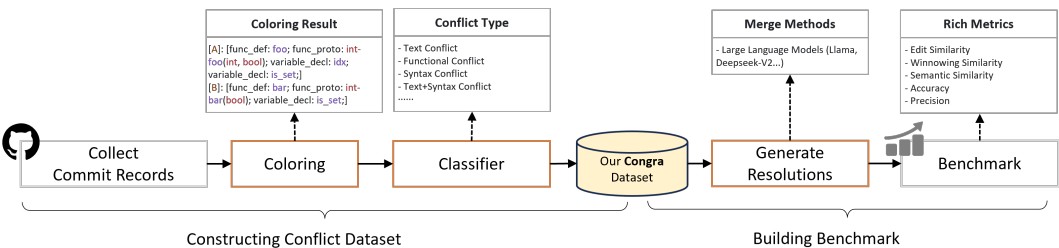

Figure 1: Benchmarking pipeline overview.

Figure 1 provides an overview of CONGRA's benchmarking pipeline. CONGRA starts with collecting open-source projects' historical merging scenarios with conflict from Github. After the raw dataset is constructed, CONGRA further colors the conflict snippets with code operations which are extracted via a lightweight but powerful syntax tree level analysis. These conflicts then are classified into various categories based on the code operations taken in the conflict code block to construct CONGRA dataset. Via the classifier, each conflict can be pigeonholed by diverse resolving complexities. Then, we combine the conflict along with its context to construct the prompt for LLMs. Finally, LLMs will respond to the query and provide conflict resolution suggestions. The results will be assessed by CONGRA's evaluation metrics.

## 3.2 COLORING

Firstly, analyzing conflict code snippets is challenging because the snippets in $M$ generated by Git are often fragmented and can contain any part or format of code. Existing program analysis tools (Lattner & Adve (2004); Wang & Shoshitaishvili (2017); Lam et al. (2011)) cannot handle these incomplete snippets syntactically or semantically. To address this, CONGRA applies $M$'s modifications from $A$ and $B$ separately to generate two merged files ($M_a$ and $M_b$). This approach applies all the difference blocks from either $A$ or $B$, allowing for code analysis of the conflicting snippets by recording the text ranges and analyzing the whole code from $M_a$ and $M_b$.

Secondly, for the benchmarking pipeline, automating code complexity assessment must use the lightest code analysis. Traditional program analysis requires a strict program context, including the project environment and necessary compilation settings, which are hard to preserve in datasets. Additionally, program analysis tools perform data flow or control flow analysis (Chess & McGraw (2004); Nielson et al. (2015)), consuming significant resources, especially for large-scale datasets. CONGRA addresses this through code operation abstraction. It extracts the operations of conflicting code fragments from $M_a$ and $M_b$ to reflect code complexity. Since CONGRA's analysis involves only lexical and grammatical analysis without deeper compilation or context dependency, it achieves efficient benchmarking.

Given $M_a$, $M_b$ and respective conflict-related code ranges, CONGRA extracts code operations from these conflict-related code blocks in the coloring phase. CONGRA first leverages tree_sitter (Brunsfeld (2024)), a lightweight multi-language code parsing framework, to convert plain text code into a syntax tree that is suitable for analysis. We pre-define types of code operations based on the node of the syntax tree. These code operations are arranged in ascending order of priority as shown below:

1. **Composite Type Definition (CTD):** Definition of composite types, e.g. the definition of class and the definition of struct in C++ language.
2. **Function Body Definition (FBD):** Definition of the function body content. Any modification to the function body is included in this operation.

3. **Function Prototype Definition (FPD):** Definition of function prototype, i.e. function name, return value type and parameter list.

4. **Language-Specific Operations (LSO):** Operations related to language features, such as macro definitions in C/C++ and the introduction of third-party libraries in Python.

5. **Commenting (CMT):** Developers' comments on the code.

6. **Variable Declaration (VD):** Declaration of all variables. These variables refers to global variables, local variables, and member variables defined in composite types.

These code operations have different priorities, and operations with higher priorities can override operations with lower priorities. For instance, the modification of a local variable name in a function body can belong to both function body definition and variable declaration. To this end, CONGRA will give priority to variable declaration as the result of code operation extraction. After analyzing each syntax tree node, CONGRA only retains code operations related to the conflicting code area, and generates code operation lists for the two versions of the code, namely $P_a$ and $P_b$. These code operations serve as the basis for conflict classification in the subsequent stage.

## 3.3 CLASSIFIER

Given $P_a$ and $P_b$, CONGRA traverses each pair of conflicting code blocks in $A$ and $B$ and classifies the conflicts according to the following conditions. For each pair of $A$'s conflict and $B$'s conflict (i.e., $P_{ai}$ in $P_a$ and $P_{bi}$ in $P_b$):

1. **Text conflict:** At least one of $P_{ai}$ and $P_{bi}$ contains CMT. Figure 2a presents a text conflict example.

2. **Functional conflict:** At least one of $P_{ai}$ and $P_{bi}$ contains FBD. Figure 2b presents a functional conflict example.

3. **Syntax conflict:** Respectively construct operations sets based on $P_{ai}$ and $P_{bi}$ (denoted as $S_a$ and $S_b$) with only remaining CTD, FPD, LSO, and VD. Finally, the difference between $S_a$ and $S_b$ is not empty. Figure 2c presents a syntax conflict example.

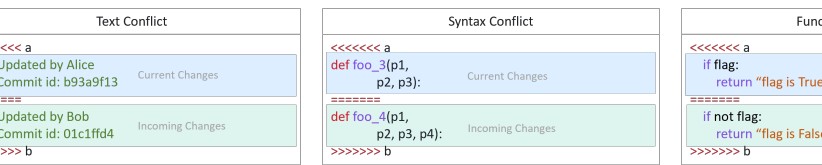

(a) Text conflict.                    (b) Syntax conflict.                    (c) Functional conflict.

Figure 2: Examples of text, syntax, and functional conflicts

Considering that the syntax among languages varies, the benchmarking of CONGRA only focuses on syntax problems caused by missing declarations or definitions which will occur in all programming languages. Since the classifier of CONGRA adopts the whitelist mode, the correctness of the classification results can be guaranteed. As one conflict may be classified into multiple categories, CONGRA supports seven classifications, i.e., all permutations of text, syntax, and functional conflict.

## 3.4 GENERATE RESOLUTIONS

In this section, we present how to obtain merge conflict resolution solutions using LLMs, as depicted in Figure 3. Here, we describe the specific steps as follows: (1) Step 1: Acquire the merge conflict and its context. Merge conflict refers to the current improvements and incoming changes. We select the previous and subsequent text at the conflict location as the context. (2) Step 2: Construct the prompt. We employ the thought-chain method to create the prompt, with the specific content provided in the Appendix A. To avoid outputting redundant context, we offer an example for LLMs to reference. (3) Step 3: Check the input. Since LLMs can only support a limited context length, we utilize the corresponding tokenizer to process the prompt. If the number of tokens surpasses the LLMs' maximum input length, revert to Step 1 and reduce the context's lines. (4) Step 4: Query

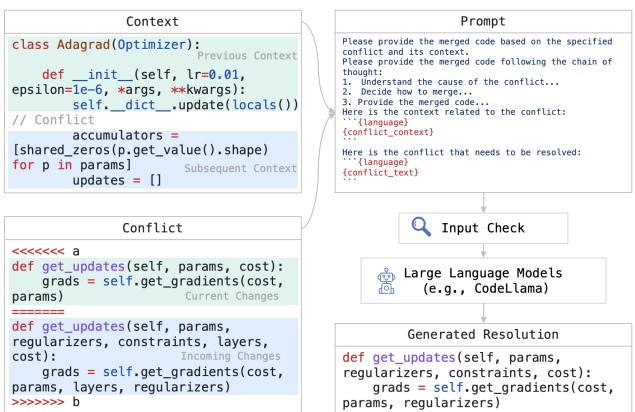

Figure 3: The process of generating resolutions by LLMs.

LLMs and obtain the resolution. We input the prompt to LLMs to receive answers, and extract the relevant code block as the final generated resolution, ensuring a logically coherent and standardized format.

## 4 GRADED CONFLICTS DATASET

We target on the well-known open-source projects in Github. Our target selection criteria is that the project should contain 10K+ lines of code and historical commits on main branch. This is because a larger code base and a more active development history can help generate conflicts with multiple complexities. Upon generating the datasets, we first traverse the historical commits of each project and reproduce each merge process through Git merge. When the merge fails, we record the conflicting file, namely $M$, and extract $A$, $B$, $O$ and $R$ from the commit graph. Finally, we utilize the coloring and classifier introduced in Section 3 to classify all the collected cases. In summary, we collected data from 34 open-source projects on GitHub: 14 written in C/C++, 11 in Java, and 9 in Python. We gathered 23,334 conflict files encompassing 44,948 conflict scenarios classified in 7 complexity types as shown in Figure 4. All these conflicts are collected with the timeframe from the initialization of these projects to 9th June 2024. The statistics on the length (in lines of code) of all conflicts and their corresponding manual resolutions are presented in Figure 5. More visualizations of CONGRA are shown in Appendix F.

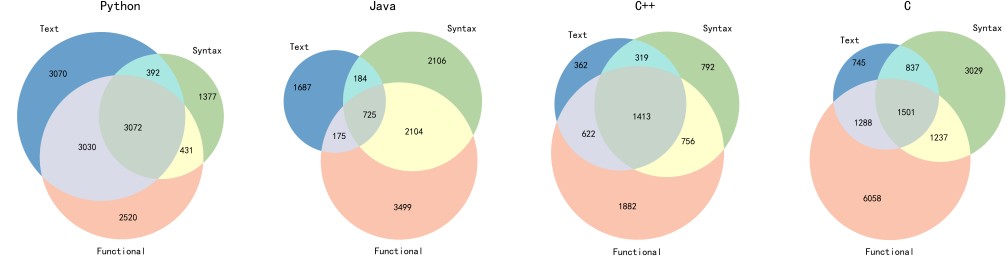

Figure 4: Venn diagrams for conflict types.

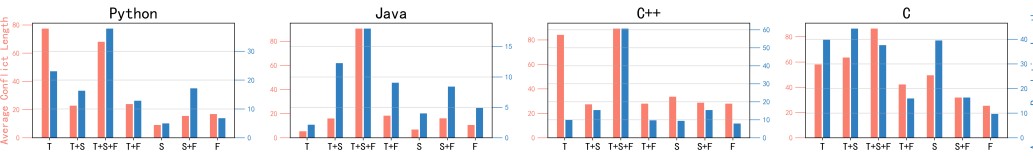

Figure 5: Average length of different types of conflicts. **T** for Text, **S** for Syntax, **F** for Functional.

## 5    BENCHMARKING FOR AUTO-CONFLICT RESOLUTOIN

### 5.1    BENCHMAKRING SETTINGS

**Evaluation Metric.** We choose the accuracy and precision of generated resolutions as the key metrics in our evaluation. Accuracy is the percentage of generated resolutions that match the ground truth to the total number of conflicts. Precision refers to the percentage of generated resolutions that match ground truth to all generated resolutions. We use the combination of normalized edit distance, winnowing, and cosine semantic similarity (Ristad & Yianilos (1998); Schleimer et al. (2003); Rahutomo et al. (2012)) as the code-matching standard (i.e., ES, WS, and SS). For a generated resolution, CONGRA regards the resolution matches the ground truth when at least one of the above values greater than 80%.

**Benchmarked Models.** As depicted in  Figure 5, CONGRA exhibits a substantial number of conflict instances, characterized by their contextual information. Consequently, we opt for LLMs that are capable of accommodating long contexts as our benchmarked models. These language models, supporting over 8K tokens, comprise three general language models (*Llama3-8B* [1], *DeepSeek-V2* [2]( DeepSeek-AI et al. (2024)), *GLM-3-turbo* [3]) and three code language models (*CodeLlama-7B*( Roziere et al. (2023)), *CodeLlama-34B*( Roziere et al. (2023)), *DeepSeek-Coder V1*( Guo et al. (2024))). To compare the performance of variants of the same model on CONGRA, we also include additional variants to form the following variant comparision group on Java and Python conflicts: (1) *DeepSeek-Coder V1* and *DeepSeek-Coder V2*( Zhu et al. (2024)). (2) *CodeLlama 7B* and *CodeLlama 34B*. (3) *LLama3-8B* and *LLama3.1-8B* [4]. We also include one of the most state-of-the-art LLM, *GPT-4o-mini* [5] as the baseline becaust it is one of the most widely-used LLMs. We only evaluate it on Java dataset because of the resource and token limitation.

**Experiment Settings.** We set the temperature coefficient uniformly at 0.7 to fully ensure the creativity of LLMs in the experiments. For each conflict, we take at most the previous 100 lines and the following 100 lines of text as context, and decrement line by line until the number of tokens is lower than the maximum context length supported by LLMs. In addition, for each conflict that does not correctly obtain the model output, we will repeat the experiment up to 10 times, otherwise it will be regarded as an unprocessable case. For open-source models, we use vLLM( Kwon et al. (2023) as the backend for model deployment.

**Comparision with Existing Merging Tool Baselines.** We initially planned to choose prior related works as the baselines to contribute to this evaluation with the selected LLMs. However, these existing tools fails to be adequate baselines because of the following reasons: (R1) For conflict dataset, although ConflictBench provides graded conflicts based on manual strategies, it does not evaluate merging tools across different conflict classifications. Since our goal is to assess LLM performance across various conflict complexities, ConflictBench's classification rules are unsuitable. Additionally, its small size (only 180 conflicts) may lead to biased results. (R2) Using program-analysis-based conflict resolving systems as benchmarks presents inconsistent metrics and unfair comparison challenges. For inconsistent metrics challenge, the aforementioned seven merging tools are either structured or semi-structured merging tools and will merge code on the AST level. As ConGra extracts merging code chunks, conflict code chunks, and resolving code chunks (i.e., the conflict resolving ground truth) on the textual level, it is impossible to map the modified zone from these merging tools to the ground truth zone, therefore fails to get the evaluation metric to compare with the LLMs' performance. For unfair comparison challenge, unlike LLMs, these systems still generate conflicts when the merging of AST nodes fails. Regarding the project scale of ConGra, more complicated merging scenarios are included in our dataset, therefore causing a high conflict ratio even in the SOTA merging tools. If we only focus on the data point without conflicts, the number of samples in the dataset will be inconsistent (compared with the number of samples used by LLMs), which causes a large deviation in the results. (R3) Regarding to machine learning approaches, RPredictor and MESTRE can predict whether the developers should keep the left version, keep the right version, or resolve the conflict manually. Unfortunately, they cannot generate any conflict

---

[1] https://llama.meta.com/llama3/

[2] We are using their official API service, which supports a context of 32K tokens instead of 128K.

[3] https://open.bigmodel.cn/dev/api#glm-3-turbo

[4] https://ai.meta.com/blog/meta-llama-3-1/

[5] https://openai.com/index/gpt-4o-mini-advancing-cost-efficient-intelligence/

resolution code therefore not suit for our benchmarking. GMerge and ChatMerge are not open-sourced thus we failed to evaluate them. DeepMerge and MergeBERT are essentially classification models that select answers from candidates, while our paper only focuses on LMs. Second, these methods are highly dependent on the training data used for fine-tuning, which has been beyond the range of our paper. Finally, these methods do not consider the contextual content and are divorced from the actual scenario, therefore cannot be used as baselines to standardize the evaluation results.

## 5.2 BENCHMARK RESULTS AND ANALYSIS

Table 2: Benchmark Result on Python and Java.

| Model | Context Length | PYTHON | | | | | JAVA | | | | |
|---|---|---|---|---|---|---|---|---|---|---|---|
| | | Accuracy | Precision | ES | WS | SS | Accuracy | Precision | ES | WS | SS |
| LLama3-8B | 8K | **75.82** | **77.45** | **0.71** | 0.36 | 0.66 | 82.93 | 83.00 | **0.75** | 0.42 | 0.67 |
| DeepSeek-V2 | 32K | 75.07 | 75.53 | 0.67 | **0.53** | **0.83** | **84.38** | **84.40** | 0.74 | **0.61** | **0.84** |
| GLM-3-turbo | 128K | 57.05 | 57.31 | 0.53 | 0.33 | 0.70 | 62.52 | 64.75 | 0.58 | 0.38 | 0.74 |
| CodeLlama-7B | 16K | 50.68 | 59.92 | 0.55 | 0.41 | 0.76 | 73.61 | 73.66 | 0.67 | 0.51 | 0.79 |
| CodeLlama-34B | 16K | 61.47 | 62.34 | 0.61 | 0.30 | 0.63 | 70.82 | 71.03 | 0.66 | 0.40 | 0.70 |
| DeepSeek-Coder V1 | 16K | 56.49 | 57.31 | 0.55 | 0.41 | 0.76 | 74.52 | 74.6 | 0.67 | 0.53 | 0.82 |

Table 3: Benchmark Result on C and C++.

| Model | Context Length | C | | | | | C++ | | | | |
|---|---|---|---|---|---|---|---|---|---|---|---|
| | | Accuracy | Precision | ES | WS | SS | Accuracy | Precision | ES | WS | SS |
| LLama3-8B | 8K | **72.45** | **73.11** | **0.64** | 0.36 | 0.69 | **78.13** | **79.22** | **0.71** | 0.38 | 0.70 |
| DeepSeek-V2 | 32K | 54.42 | 71.06 | 0.61 | **0.45** | 0.79 | 70.86 | 77.31 | 0.69 | **0.53** | **0.83** |
| GLM-3-turbo | 128K | 58.86 | 60.32 | 0.52 | 0.30 | 0.70 | 64.10 | 64.13 | 0.57 | 0.32 | 0.70 |
| CodeLlama-7B | 16K | 59.09 | 60.55 | 0.52 | 0.35 | 0.73 | 64.28 | 64.9 | 0.58 | 0.38 | 0.75 |
| CodeLlama-34B | 16K | 67.75 | 68.23 | 0.61 | 0.26 | 0.63 | 69.83 | 70.39 | 0.66 | 0.29 | 0.63 |
| DeepSeek-Coder V1 | 16K | 62.36 | 62.72 | 0.54 | 0.38 | **0.77** | 62.33 | 62.9 | 0.57 | 0.39 | 0.78 |

### 5.2.1 OVERALL PERFORMANCE

In this section, we present a comprehensive evaluation of advanced LLMs on our proposed CONGRA dataset. The experimental results can be found in Tables 2 and 3. Upon careful comparison, we draw the following observations: (1) LLMs with longer context support do not always yield optimal results. GLM-3-Turbo, which supports a 128K context, is generally outperformed by LLama3-8B, which only supports an 8K context, with the exception of semantic similarity. For instance, in Python and Java, the Precision metric for GLM-3-Turbo is nearly 20% lower than that of LLama3-8B. We speculate that this may result from LLMs with ultra-long context support not being sufficiently trained on merge conflict datasets, hindering their ability to effectively extract valuable information from the context to suggest merge resolutions. (2) Code LLMs do not appear to demonstrate a distinct advantage. Notably, LLama3-8B and DeepSeek-V2 surpass Code LLMs in Precision across all four languages. We attribute this to two factors: (i) general models are in fact trained on extensive code repositories, so their code comprehension capabilities are not significantly inferior to Code LLMs. (ii) Elements within conflicts such as comments and variable names contribute a plethora of semantic information, extending beyond mere code understanding, which presents a challenge for Code LLMs. (3) Both LLama3-8B and DeepSeek-V2 prove to be well-suited for automatic conflict resolution. We also conducted experiments to evaluate the performance of different variants of models on CONGRA in Appendix G, and explore the most state-of-the-art models' performance on CONGRA in Appendix H.

### 5.2.2 THE IMPACT OF CONFLICT TYPE

Furthermore, we investigate the performance of LLMs under different conflict types. Based on Section 3.3, we consider seven conflict types, namely Text, Syntax, Functional conflicts, and their combinations. The results are visualized in Figure 6. In Figure 6, the closer the result is to blue, the better the model performs; the closer it is to green, the worse the model performs.

For LLMs, a simpler conflict does not necessarily mean it is easier to handle. This is actually a counterintuitive phenomenon. Specifically, for most models and languages (e.g., Python, C/C++), LLMs exhibit better performance on the most complex conflicts (i.e., F+S+T) and poorer performance

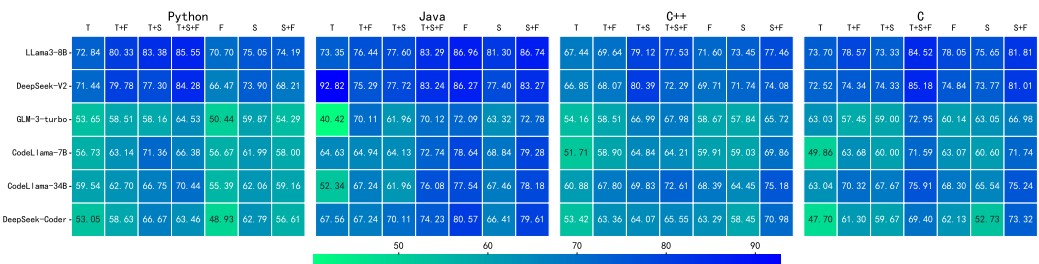

Figure 6: Heatmap on precision of LLMs and types. **T** for Text, **S** for Syntax, **F** for Functional.

on the simplest conflicts (i.e., F, S, and T). In the context of text conflicts, the performance of LLMs on different samples can be ranked as follows: T+F+S > T+F ≈ T+S > T. We believe that this occurs because, for LLMs, the simpler the samples within the conflict area (such as type T), the less effective guidance information the conflict can provide, causing LLMs to produce conservative answers. For example, when the conflict involves a comment, LLMs tend to preserve as much information as possible from both branches a and b. In contrast, the more complex the conflict area, the clearer the direction for conflict resolution. In summary, LLMs need to extract more valuable guidance information from the conflict area itself to improve their performance. Notice that in Figure 6, the precision of F in Java is relatively higher. This is due to Java's encapsulation. To hide "sensitive" member variables of a class from users, these variables are often declared as private, and the class must provide public *get* and *set* methods to access and update these values. Consequently, these getter and setter methods are simple in terms of functionality. Since most of the F conflicts in the Java dataset arise from the rewriting of these encapsulated methods, the LLMs can easily infer the resolutions, resulting in high precision.

## 5.3 THE ROLE OF CONTEXT

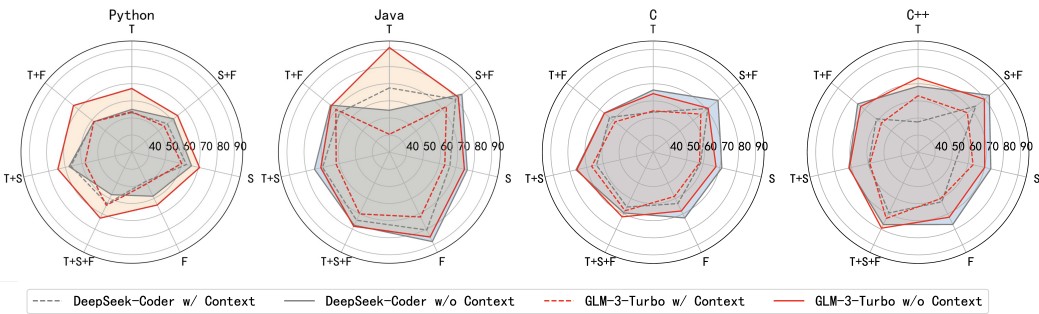

Figure 7: The impact of contextual information on Precision.

Table 4: Benchmark Result for Different Context Lines on Java.

| Model | # Context Line | JAVA | | | | |
|---|---|---|---|---|---|---|
| | | Accuracy | Precision | ES | WS | SS |
| DeepSeer-Coder V2 | 20 | 86.08 | 86.13 | **0.76** | 0.63 | **0.85** |
| | 50 | **86.26** | **86.31** | **0.76** | **0.64** | **0.85** |
| | 100 | 85.89 | 86.20 | **0.76** | **0.64** | **0.85** |
| LLama3.1-8B | 20 | 78.07 | 78.13 | 0.70 | 0.55 | 0.83 |
| | 50 | 79.10 | 79.16 | 0.71 | 0.56 | 0.83 |
| | 100 | **81.01** | **81.08** | **0.72** | **0.58** | **0.84** |

In the vast majority of cases, LLMs without conflict context information significantly outperform those with context information. As observed in Figure 7, the red and gray solid circles almost entirely encompass their corresponding dashed circles, with only a few exceptions. We analyze

this phenomenon from two perspectives: (1) the context understanding capability of large language models. Although existing LLMs have undergone extensive training on code data, this does not necessarily imply that they can effectively extract useful information from the rich context for automatic conflict resolution. (2) The choice of context. In our experiments, we provided the most basic context information (i.e., the adjacent codes around the current conflict chunk). However, regarding the actual programming process, a session of code relies more on variables, functions, and composite types that are usually defined far away from where the current code chunk is located, or even in another file that the aforementioned concept of basic context cannot cover. By the same token, using these invalid contexts is possible to introduce significant amounts of noise and can cause LLM to lose focus, which in turn leads to worse performance. As a result, selecting the appropriate context for automatic conflict resolution may be a valuable research direction to explore. We conducted additional experiment with various context lines (depicted in Table 4) to evaluate the conflict resolution ability of *DeepSeeker-Coder V2* and *LLama3.1-8B* as illustration. We change the context lines number to 20, 50 and 100 respectively and compare the five metrics as well. According to the result, although a larger number of context lines leads to better conflict resolving performance in *LLama3.1-8B*, a worse performance is found in *DeepSeek-Code V2*.

## 6 DISCUSSION

**Limitation.** The conflict classification strategy in Section 3 ensures correct classification but may fail to identify the exact category of a conflict due to advanced language usage. For example, the use of template class definitions in C++ (Vandevoorde & Josuttis (2002)) prevents tree_sitter (Brunsfeld (2024)) from capturing the type used as the template argument during instantiation. Consequently, a conflict that should be classified as "syntax" may be regarded as "non-syntax." We will continue to refine CONGRA's classification in future work.

**Societal Impacts.** (1) Improvement of benchmarking current LMs on code merging tasks. During the merging process, LMs may fail to handle pure text conflict, and generate error-prone merged code with incorrect syntax or functionality. These wrong results require large amounts of human intervention to fix. With ConGra, these LMs can be well benchmarked in terms of their ability to generate text-correct, syntactic-correct, and functional-correct merged code. We believe the publication of ConGra can further enhance the utilization of LMs in the code merging field and cast a positive impact on the whole community. (1) Propagation of large datasets for LMs' training and testing. There is a trend that more large-scale Git repository data are used for LLMs purposes. We do believe that the publication of ours or similar works will fuel the momentum as well. We think the open-source of large datasets will finally lead to a positive contribution..

## 7 CONCLUSION

We propose CONGRA, a complexity-graded conflict benchmarking system. CONGRA implements a highly efficient and accurate conflict classification algorithm to construct a complexity-graded conflict dataset, which is used to evaluate the performance of merging tools under various conflict scenarios. CONGRA utilizes three code matching metrics of different granularities and combines them to calculate the accuracy and precision of auto-generated resolutions. We evaluate six LLMs on CONGRA, and the results show that LLMs with longer context support often perform worse than those with shorter context support, and general LLMs outperform specialized code LLMs in precision.

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

## A  PROMPT

```
[System]
You are an expert in code merge conflicts, providing the merged code based on the conflict and its context.

[User]
Please provide the merged code based on the specified conflict and its context.
Please provide the merged code following the chain of thought:
1. Understand the cause of the conflict: Examine the conflicting code and its context to understand why the
conflict occurred.
2. Decide how to merge: Based on the functionality and logic of the code, determine which changes should be
kept or how the changes from both sides can be combined.
3. Provide the merged code, using "```{language}" as the beginning and "```" as the end of the merged code.
You only need to output the resolution of the conflict without providing any context.
For example,
Conflict Context is:
```python
def quick_sort(arr):
    <<<<<<< a
    if len(arr) <= 1:
        return arr
    else:
        pivot = arr[0]
        left = [x for x in arr[1:] if x < pivot]
        right = [x for x in arr[1:] if x >= pivot]
        return quick_sort(left) + [pivot] + quick_sort(right)
    =======
    n = len(arr)
    for i in range(n):
        for j in range(0, n-i-1):
            if arr[j] > arr[j+1] :
                arr[j], arr[j+1] = arr[j+1], arr[j]
    >>>>>>> b
    return arr
```
Conflict is:
```python
    <<<<<<< a
    if len(arr) <= 1:
        return arr
    else:
        pivot = arr[0]
        left = [x for x in arr[1:] if x < pivot]
        right = [x for x in arr[1:] if x >= pivot]
        return quick_sort(left) + [pivot] + quick_sort(right)
    =======
    n = len(arr)
    for i in range(n):
        for j in range(0, n-i-1):
            if arr[j] > arr[j+1] :
                arr[j], arr[j+1] = arr[j+1], arr[j]
    >>>>>>> b
```
You need to output:
```python
    if len(arr) <= 1:
        return arr
    else:
        pivot = arr[0]
        left = [x for x in arr[1:] if x < pivot]
        right = [x for x in arr[1:] if x >= pivot]
    return quick_sort(left) + [pivot] + quick_sort(right)
```
Here is the context related to the conflict:
```{language}
{conflict_context}
```
Here is the conflict that needs to be resolved:
```{language}
{conflict_text}
```
```

Figure 8: Prompt

## B  DEMO CASE

### B.1  CONFLICT

Listing 1: Demo 1: text conflict

```
1  <<<<<<< a
2          padding: Int, or tuple of 3 ints, or tuple of 3 tuples of 2 ints.
```

```
 3              - If int: the same symmetric padding is applied to depth, height,
 4                and width.
 5              - If tuple of 3 ints: interpreted as three different symmetric
 6                padding values for depth, height, and width:
 7                `(symmetric_dim1_pad, symmetric_dim2_pad, symmetric_dim3_pad)`.
 8              - If tuple of 3 tuples of 2 ints: interpreted as
 9                `((left_dim1_pad, right_dim1_pad), (left_dim2_pad,
10                right_dim2_pad), (left_dim3_pad, right_dim3_pad))`.
11          data_format: A string, one of `"channels_last"` (default) or
12              `"channels_first"`. The ordering of the dimensions in the inputs.
13              `"channels_last"` corresponds to inputs with shape
14              `(batch_size, spatial_dim1, spatial_dim2, spatial_dim3, channels)`
15              while `"channels_first"` corresponds to inputs with shape
16              `(batch_size, channels, spatial_dim1, spatial_dim2, spatial_dim3)`.
17              When unspecified, uses `image_data_format` value found in your Keras
18              config file at `~/.keras/keras.json` (if exists). Defaults to
19              `"channels_last"`.
20 ======
21      padding: Int, or tuple of 3 ints, or tuple of 3 tuples of 2 ints.
22          - If int: the same symmetric padding
23            is applied to height and width.
24          - If tuple of 3 ints:
25            interpreted as two different
26            symmetric padding values for height and width:
27            `(symmetric_dim1_pad, symmetric_dim2_pad, symmetric_dim3_pad)`.
28          - If tuple of 3 tuples of 2 ints:
29            interpreted as
30            `((left_dim1_pad, right_dim1_pad), (left_dim2_pad,
31              right_dim2_pad), (left_dim3_pad, right_dim3_pad))`
32      data_format: A string,
33          one of `channels_last` (default) or `channels_first`.
34          The ordering of the dimensions in the inputs.
35          `channels_last` corresponds to inputs with shape
36          `(batch_size, spatial_dim1, spatial_dim2, spatial_dim3, channels)`
37          while `channels_first` corresponds to inputs with shape
38          `(batch_size, channels, spatial_dim1, spatial_dim2, spatial_dim3)`.
39          It defaults to the `image_data_format` value found in your
40          Keras config file at `~/.keras/keras.json`.
41          If you never set it, then it will be "channels_last".
42 >>>>>>> b
```

Listing 2: Demo 2: text and functional conflict

```
 1 <<<<<<< a
 2     constraints = _process_dynamic_shapes(mod, args, kwargs, dynamic_shapes) or []
 3
 4     kwargs = kwargs or {}
 5 ======
 6     if constraints is not None:
 7         log_export_usage(event="export.private_api", flags={"constraints"})
 8         warnings.warn(
 9             "Using `constraints` to specify dynamic shapes for export is DEPRECATED "
10             "and will not be supported in the future. "
11             "Please use `dynamic_shapes` instead (see docs on `torch.export.export`).",
12             DeprecationWarning,
13             stacklevel=2,
14         )
15     else:
16         constraints = _process_dynamic_shapes(f, args, kwargs, dynamic_shapes) or []
17 >>>>>>> b
```

Listing 3: Demo 3: syntax conflict

```
 1 <<<<<<< a
 2 import torch.utils._pytree as pytree
 3 ======
 4 >>>>>>> b
 5 from torch._decomp import register_decomposition
```

Listing 4: Demo 4: text, syntax, and functional conflict

```
 1 <<<<<<< a
 2             log.info("converting frame raised error, suppressing error")
 3 ======
 4
 5             # Suppress the error.  NB: It's very important to do the
```

```
 6                  # suppression logging HERE, where the actual suppression
 7                  # happens. Previously it was somewhere else and so it was
 8                  # possible to accidentally not log at all.
 9                  record_filename = getattr(e, "record_filename", None)
10                  code = frame.f_code
11                  if config.is_fbcode():
12                      from torch._dynamo.fb.logging import (  # type: ignore[import]
13                          log_dynamo_suppress_errors,
14                      )
15
16                      error_msg = format_error_msg_verbose(e, code, record_filename, frame)
17                      log_dynamo_suppress_errors(
18                          code.co_name, code.co_filename, code.co_firstlineno, error_msg
19                      )
20                  else:
21                      error_msg = format_error_msg(e, code, record_filename, frame)
22
23                  if soft_fail:
24                      log.info(error_msg, exc_info=True)
25                  else:
26                      log.warning(error_msg, exc_info=True)
27 >>>>>>> b
28          return None
```

## B.2 GROUND TRUTH

### Listing 5: Ground truth of demo 1

```
 1      Args:
 2          padding: Int, or tuple of 3 ints, or tuple of 3 tuples of 2 ints.
 3              - If int: the same symmetric padding is applied to depth, height,
 4                and width.
 5              - If tuple of 3 ints: interpreted as three different symmetric
 6                padding values for depth, height, and width:
 7                `(symmetric_dim1_pad, symmetric_dim2_pad, symmetric_dim3_pad)`.
 8              - If tuple of 3 tuples of 2 ints: interpreted as
 9                `((left_dim1_pad, right_dim1_pad), (left_dim2_pad,
10                right_dim2_pad), (left_dim3_pad, right_dim3_pad))`.
11          data_format: A string, one of `"channels_last"` (default) or
12              `"channels_first"`. The ordering of the dimensions in the inputs.
13              `"channels_last"` corresponds to inputs with shape
14              `(batch_size, spatial_dim1, spatial_dim2, spatial_dim3, channels)`
15              while `"channels_first"` corresponds to inputs with shape
16              `(batch_size, channels, spatial_dim1, spatial_dim2, spatial_dim3)`.
17              When unspecified, uses `image_data_format` value found in your Keras
18              config file at `~/.keras/keras.json` (if exists). Defaults to
19              `"channels_last"`.
```

### Listing 6: Ground truth of demo 2

```
 1
 2      kwargs = kwargs or {}
 3      _process_dynamic_shapes(mod, args, kwargs, dynamic_shapes)  # TODO(avik): remove
```

### Listing 7: Ground truth of demo 3

```
 1 import torch._inductor as inductor
 2 import torch.utils._pytree as pytree
 3 from torch import fx
 4 from torch._decomp import register_decomposition
```

### Listing 8: Ground truth of demo 4

```
 1                  raise
 2
 3                  # Suppress the error.  NB: It's very important to do the
 4                  # suppression logging HERE, where the actual suppression
 5                  # happens. Previously it was somewhere else and so it was
 6                  # possible to accidentally not log at all.
 7                  record_filename = getattr(e, "record_filename", None)
 8                  code = frame.f_code
 9                  error_msg = format_error_msg(e, code, record_filename, frame)
10
```

```
11              if soft_fail:
12                  log.info(error_msg, exc_info=True)
13              else:
14                  log.warning(error_msg, exc_info=True)
```

## B.3 RESOLUTION GENERATED BY LLAMA3-8B

Listing 9: Demo 1: LLama3-8B resolution

```
1      padding: Int, or tuple of 3 ints, or tuple of 3 tuples of 2 ints.
2          - If int: the same symmetric padding is applied to depth, height, and width.
3          - If tuple of 3 ints: interpreted as three different symmetric padding values for depth
             , height, and width:
4            `(symmetric_dim1_pad, symmetric_dim2_pad, symmetric_dim3_pad)`.
5          - If tuple of 3 tuples of 2 ints: interpreted as
6            `((left_dim1_pad, right_dim1_pad), (left_dim2_pad, right_dim2_pad), (left_dim3_pad,
                right_dim3_pad))`.
7      data_format: A string, one of `"channels_last"` (default) or
8          `"channels_first"`. The ordering of the dimensions in the inputs.
9          `"channels_last"` corresponds to inputs with shape
10         `(batch_size, spatial_dim1, spatial_dim2, spatial_dim3, channels)`
11         while `"channels_first"` corresponds to inputs with shape
12         `(batch_size, channels, spatial_dim1, spatial_dim2, spatial_dim3)`.
13         When unspecified, uses `image_data_format` value found in your Keras
14         config file at `~/.keras/keras.json` (if exists). Defaults to
15         `"channels_last"`.
```

Listing 10: Demo 2: LLama3-8B resolution

```
1      if constraints is not None:
2          log_export_usage(event="export.private_api", flags={"constraints"})
3          warnings.warn(
4              "Using `constraints` to specify dynamic shapes for export is DEPRECATED "
5              "and will not be supported in the future. "
6              "Please use `dynamic_shapes` instead (see docs on `torch.export.export`).",
7              DeprecationWarning,
8              stacklevel=2,
9          )
10     else:
11         constraints = _process_dynamic_shapes(f, args, kwargs, dynamic_shapes) or []
12     kwargs = kwargs or {}
```

Listing 11: Demo 3: LLama3-8B resolution

```
1  from torch.utils._pytree import pytree
```

Listing 12: Demo 4: LLama3-8B resolution

```
1          log.info("converting frame raised error, suppressing error")
```

## B.4 RESOLUTION GENERATED BY DEEPSEEK-CHAT

Listing 13: Demo 1: Deepseek-Chat resolution

```
1      padding: Int, or tuple of 3 ints, or tuple of 3 tuples of 2 ints.
2          - If int: the same symmetric padding is applied to depth, height,
3            and width.
4          - If tuple of 3 ints: interpreted as three different symmetric
5            padding values for depth, height, and width:
6            `(symmetric_dim1_pad, symmetric_dim2_pad, symmetric_dim3_pad)`.
7          - If tuple of 3 tuples of 2 ints: interpreted as
8            `((left_dim1_pad, right_dim1_pad), (left_dim2_pad,
9            right_dim2_pad), (left_dim3_pad, right_dim3_pad))`.
10     data_format: A string, one of `"channels_last"` (default) or
11         `"channels_first"`. The ordering of the dimensions in the inputs.
12         `"channels_last"` corresponds to inputs with shape
13         `(batch_size, spatial_dim1, spatial_dim2, spatial_dim3, channels)`
14         while `"channels_first"` corresponds to inputs with shape
15         `(batch_size, channels, spatial_dim1, spatial_dim2, spatial_dim3)`.
16         It defaults to the `image_data_format` value found in your
```

```
17              Keras config file at `~/.keras/keras.json`.
18              If you never set it, then it will be "channels_last".
```

Listing 14: Demo 2: Deepseek-Chat resolution

```python
1   if constraints is not None:
2       log_export_usage(event="export.private_api", flags={"constraints"})
3       warnings.warn(
4           "Using `constraints` to specify dynamic_shapes_for_export_is_DEPRECATED_"
5           "and_will_not_be_supported_in_the_future._"
6           "Please_use_`dynamic_shapes`_instead_(see_docs_on_`torch.export.export`).",
7           DeprecationWarning,
8           stacklevel=2,
9       )
10  else:
11      constraints = _process_dynamic_shapes(f, args, kwargs, dynamic_shapes) or []
```

Listing 15: Demo 3: Deepseek-Chat resolution

```python
1   import torch.utils._pytree as pytree
2   from torch._decomp import register_decomposition
```

Listing 16: Demo 4: Deepseek-Chat resolution

```python
1               # Suppress the error.  NB: It's very important to do the
2               # suppression logging HERE, where the actual suppression
3               # happens. Previously it was somewhere else and so it was
4               # possible to accidentally not log at all.
5               record_filename = getattr(e, "record_filename", None)
6               code = frame.f_code
7               if config.is_fbcode():
8                   from torch._dynamo.fb.logging import (  # type: ignore[import]
9                       log_dynamo_suppress_errors,
10                  )
11
12                  error_msg = format_error_msg_verbose(e, code, record_filename, frame)
13                  log_dynamo_suppress_errors(
14                      code.co_name, code.co_filename, code.co_firstlineno, error_msg
15                  )
16              else:
17                  error_msg = format_error_msg(e, code, record_filename, frame)
18
19              if soft_fail:
20                  log.info(error_msg, exc_info=True)
21              else:
22                  log.warning(error_msg, exc_info=True)
23          return None
```

## B.5 RESOLUTION GENERATED BY DEEPSEEK-CODER

Listing 17: Demo 1: Deepseek-Coder resolution

```
1   padding: Int, or tuple of 3 ints, or tuple of 3 tuples of 2 ints.
2       - If int: the same symmetric padding is applied to depth, height,
3         and width.
4       - If tuple of 3 ints: interpreted as three different symmetric
5         padding values for depth, height, and width:
6         `(symmetric_dim1_pad, symmetric_dim2_pad, symmetric_dim3_pad)`.
7       - If tuple of 3 tuples of 2 ints: interpreted as
8         `((left_dim1_pad, right_dim1_pad), (left_dim2_pad,
9           right_dim2_pad), (left_dim3_pad, right_dim3_pad))`.
10  data_format: A string, one of `"channels_last"` (default) or
11      `"channels_first"`. The ordering of the dimensions in the inputs.
12      `"channels_last"` corresponds to inputs with shape
13      `(batch_size, spatial_dim1, spatial_dim2, spatial_dim3, channels)`
14      while `"channels_first"` corresponds to inputs with shape
15      `(batch_size, channels, spatial_dim1, spatial_dim2, spatial_dim3)`.
16      When unspecified, uses `image_data_format` value found in your Keras
17      config file at `~/.keras/keras.json` (if exists). Defaults to
18      `"channels_last"`.
```

Listing 18: Demo 2: Deepseek-Coder resolution

```
1    constraints = _process_dynamic_shapes(f, args, kwargs, dynamic_shapes) or []
2
3    kwargs = kwargs or {}
```

Listing 19: Demo 3: Deepseek-Coder resolution

```
1  import torch.utils._pytree as pytree
2  from torch._decomp import register_decomposition
```

Listing 20: Demo 4: Deepseek-Coder resolution

```
1   log.info("converting_frame_raised_error,_suppressing_error")
2   record_filename = getattr(e, "record_filename", None)
3   code = frame.f_code
4   if config.is_fbcode():
5       from torch._dynamo.fb.logging import (  # type: ignore[import]
6           log_dynamo_suppress_errors,
7       )
8
9       error_msg = format_error_msg_verbose(e, code, record_filename, frame)
10      log_dynamo_suppress_errors(
11          code.co_name, code.co_filename, code.co_firstlineno, error_msg
12      )
13  else:
14      error_msg = format_error_msg(e, code, record_filename, frame)
15
16  if soft_fail:
17      log.info(error_msg, exc_info=True)
18  else:
19      log.warning(error_msg, exc_info=True)
20  return None
```

## C  GIT MERGE CONFLICT EXAMPLE

Figure 9 shows a code merging scenario with conflict. Both version A (Figure 9a) and version B ( Figure 9b) implement quick sort, but version A selects the middle element of *arr* as the pivot, while version B selects the first element. This discrepancy causes Git to encounter an impasse and report a conflict, as depicted in  Figure 9c.

(a) Quick sort from version A.   (b) Quick sort from version B.   (c) Quick sort with merge conflict.

Figure 9: Quick sort merging example

## D  LICENSE

We utilized the source code of 34 open-source projects in this research. These projects and their license information are as follows: 1) Linux Kernel, Android Kernel, Raspberry Pi Kernel, Git, MySQL, ReactOS, JDK, NewPipe, Ansible Cpython, are licensed under GNU General Public License (GPL); 2) Bitcoin, GCC, Jenkins, are licensed under the Massachusetts Institute of Technology (MIT); 3) LLVM, Swift, Tensorflow, AOSP, dbeaver, Ghidra, hadoop, Micronaut, Netty, Sprint-boot, Sprint-framework, Keras, Transformers, are under the Apache License; 4) Mongo is under Server Side Public License; 5) PHP is under PHP License; 6) V8, Django, Pandas, Pytorch, Scrapy, are under BSD-3-Clause License; 7) Eclipse is under Eclipse Public License; 8) Youtube-dl is unlicensed. We acknowledge the contributions of the open-source community in developing and maintaining these projects.

## E    CODE SIMILARITY METRICS

- **Edit Similarity (ES)**. Edit distance is a metric used to measure the difference between two strings (Ristad & Yianilos (1998)). It is defined as the minimum number of edit operations required to transform one string into another. In CONGRA, we normalize the edit distance, treating the normalized result as the similarity measure in character level.
- **Winnowing Similarity (WS)**. Winnowing is an algorithm used for text similarity detection and fingerprint extraction (Schleimer et al. (2003)). It has been implemented in the Moss code plagiarism detection system (Aiken (2022)) to determine code similarity. In CONGRA, we normalize the winnowing result to assess the similarity of the whole conflict code snippets.
- **Semantic Similarity (SS)**. We use the cosine similarity between the generated resolution and the actual resolution as the semantic similarity. In CONGRA, in order to effectively model the semantic information of the resolution, we use `BCEmbedding` NetEase Youdao (2023) as the embedding model to obtain the representation of each resolution.

## F    VISUALIZATION OF THE NUMBER OF CONFLICTS IN CONGRA

Here we visualize the distribution characteristics of the number of conflicts contained in each file in four languages in Figure 10. Overall, most files contain only one or two conflicts, and a very small number of files contain more than 10 conflicts. We also visualize the number of different types of conflicts for each language in Figure 11.

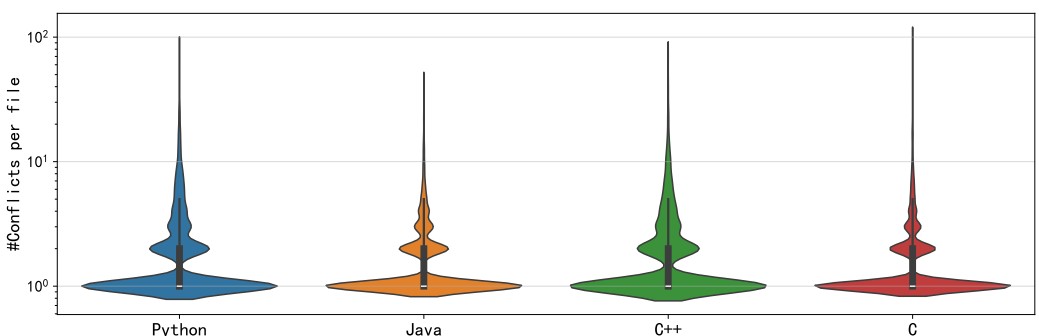

Figure 10: Violin plots of the number of conflicts per file.

## G    COMPARISON OF MODEL VARIANTS ON CONGRA

Table 5: Benchmark Result for Model Variants on Python and Java.

| Model | Context Length | PYTHON | | | | | JAVA | | | | |
|---|---|---|---|---|---|---|---|---|---|---|---|
| | | Accuracy | Precision | ES | WS | SS | Accuracy | Precision | ES | WS | SS |
| LLama3-8B | 8K | **75.82** | **77.45** | **0.71** | 0.36 | 0.66 | **82.93** | **83.00** | **0.75** | 0.42 | 0.67 |
| LLama3.1-8B | 8k | 73.58 | 75.16 | 0.67 | **0.53** | **0.83** | 81.01 | 81.08 | 0.72 | **0.58** | **0.84** |
| CodeLlama-7B | 16K | 50.68 | 59.92 | 0.55 | **0.41** | **0.76** | **73.61** | **73.66** | **0.67** | **0.51** | **0.79** |
| CodeLlama-34B | 16K | **61.47** | **62.34** | **0.61** | 0.30 | 0.63 | 70.82 | 71.03 | 0.66 | 0.40 | 0.70 |
| DeepSeek-Coder V1 | 16K | 56.49 | 57.31 | 0.55 | 0.41 | 0.76 | 74.52 | 74.6 | 0.67 | 0.53 | 0.82 |
| DeepSeek-Coder V2 | 16k | **77.04** | **78.14** | **0.70** | **0.56** | **0.84** | **85.89** | **86.20** | **0.76** | **0.64** | **0.85** |

We evaluated the performance of different variants of LLMs with the following comparison groups:

1. *DeepSeek-Coder V1* vs *DeepSeek-Coder V2*.
2. *CodeLlama 7B* vs *CodeLlama 34B*.
3. *LLama3-8B* vs *LLama3.1-8B*.

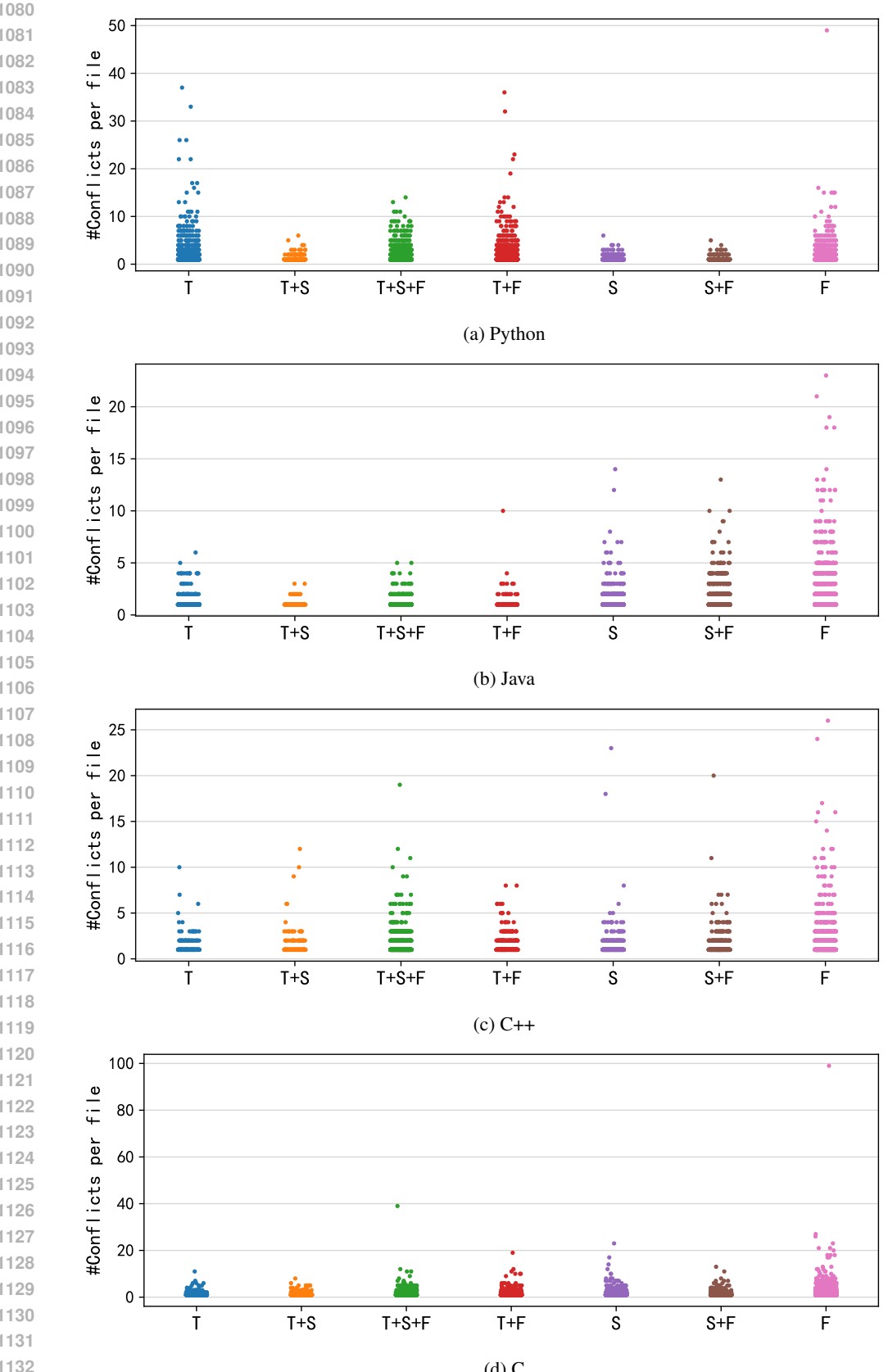

Figure 11: Strip plots of number of conflicts per file.

Table 5 shows the results of the variants comparision. In Conclusion, *DeepSeek-Coder V2* gains better performance on all of the five metrics in both Python and Java targets compared with *DeepSeek-Coder V1*. *LLama3-8B* outperforms *LLama3.1-8B* in terms of Accuracy, Precision, and ES while *LLama3.1-8B* takes the lead in WS and SS regarding both Python and Java projects. *CodeLlama-7B* performs better on all of the five metrics in Java datasets but worse on Accuracy, Precision, and ES in Python datasets.

## H    COMPARISON AMONG STATE-OF-THE-ART MODELS ON CONGRA

Table 6: Benchmark Result for State-of-the-art baseline and Well-performed Models on Java.

| Model | JAVA | | | | |
|---|---|---|---|---|---|
| | Accuracy | Precision | ES | WS | SS |
| DeepSeer-Coder V2 | **86.08** | **86.13** | **0.76** | **0.63** | **0.85** |
| LLama3.1-8B | 78.07 | 78.13 | 0.70 | 0.55 | 0.83 |
| GPT-4o-mini | 83.76 | 83.82 | 0.74 | 0.62 | 0.84 |

To explore whether CONGRA can help with the improvement of the state-of-the-art LMs, we conducted additional experiments as shown in Table 6. Due to the resource and token limitations, we restrict the number of context lines to 20. We include *GPT-4o-mini*, *DeepSeek-Coder V2* and *LLama3.1-8B* as they are either one of the most popular LMs or performed outstandingly in the other parts of our evaluation. Notably, DeepSeek-Coder V2 performed the best among the three SOTA LLMs. Nevertheless, DeepSeek-Coder V2 does not exhibit an incredibly high performance (1.70%↑ in Accuracy, 1.73%↑ in Precision, 0.01↑ in ES, 0.02↑ in WS, and 0.01↑ in SS). To this end, we suggest that there is still room for improvement of LLMs on code merging tasks.

