# OpenReview forum: "ConGra: Benchmarking Automatic Conflict Resolution"
_ICLR.cc/2025/Conference — ICLR 2025 Conference Withdrawn Submission_

### Official Review · Reviewer_dgxq · 2024-10-29

**Soundness:** 2
**Presentation:** 3
**Contribution:** 2
**Rating:** 5
**Confidence:** 3

**Summary:**

This paper proposes ConGra, a large-scale benchmark for evaluating LLMs on solving conflicts when merging different software versions. The authors built the ConGra dataset in 3 steps: collect Github commits, coloring by program analysis, and classify the different types of conflicts. Then, the authors propose a general pipeline for using LLMs to generate merged code. This paper evaluates 6 open-source LLMs on the ConGra dataset, and reaches several findings: first, LLMs with longer contexts do not always perform better than models with shorter contexts. Second, general-purpose LLMs outperform code LLMs in resolution precision and accuracy.

**Strengths:**

+: This paper first investigates the performance of LLMs in resolving software version conflicts.

+: Proposed a conflict dataset in multiple programming languages.

**Weaknesses:**

-: The authors only adopt a few models for evaluation, which makes their conclusion less convincing. For example, it is better to include results for more proprietary models, such as GPT-4o.

-: The authors stated that GPT-4o-mini is only evaluated on the Java subset due to its token limitation, but the context window of GPT-4o-mini is 128K, which is comparable to other models in the evaluation.

-: The finding "general-purpose LLMs outperform code LLMs" may be inaccurate. The code LLMs in this paper, CodeLlama and DeepSeek-Coder, are fine-tuned on older versions of base models than Llama-3 and DeepSeek-V2. To make a fair comparison, the authors should compare these code LLMs with their corresponding base models.

**Questions:**

Please see "weaknesses".

---

### Official Review · Reviewer_MBtc · 2024-11-01

**Soundness:** 3
**Presentation:** 2
**Contribution:** 2
**Rating:** 3
**Confidence:** 5

**Summary:**

The paper presents CONGRA, a benchmark for program merge conflict resolution. The CONGRA dataset  includes 44,948 merge conflict cases and resolutions collected from 34 real-world projects spanning multiple programming languages. This dataset classifies conflicts into seven types, enabling more granular evaluations.

The benchmark uses edit distance, winnowing, and cosine similarity to measure the accuracy and precision of conflict resolution results. Authors evaluate six LLMs on the CONGRA benchmark, finding that general-purpose LLMs frequently outperform code-specialized ones in resolving code conflicts and that the models with shorter context windows sometimes achieve better results than those with longer context windows.

**Strengths:**

1) The paper tackles an important software engineering problem -- automated merge conflict resolution
2) The effort promotes open science, by promising to open source the benchmark which could help drive the research in automated program merge

**Weaknesses:**

The benchmark matching criteria are not optimal, and are not well aligned with the evaluation metrics used in the existing merge conflict research. It appears to be more optimized towards a typical code in-filling task. More specifically, CONGRA regards the resolution candidate matching the ground truth when similarity is greater than 80%. To my opinion, this is a major issue and I see two possible solutions:
1) require a stricter 100% syntactic match (modulo the white space and indentation). A variant of this could also exclude local identifiers defined and used only in the conflict region scope from the similarity calculation, given the freedom in choosing the particular identifier names in this case. Moreover, there is no clear reason to choose 80% similarity threshold -- the numbers in the table 2-3 are already quite high suggesting the benchmark maybe too easy and is at risk of becoming obsolete in the next year or so.
2) better option: verify functional correctness instead. When constructing the benchmark you could select only the conflicts that have unit test coverage for the lines in the conflicted region in the ground truth user-resolved version of the code. In this case, the functional correctness would be verified wrt to the unit tests

**Questions:**

Classification of Merge Conflicts: The current classification of merge conflicts feels too restrictive and maybe disconnected from the merge task, appropriate for broader infilling tasks -- what portion of real-world merge conflicts would it cover? I believe the classification should reflect the difficulty of resolution. From this perspective, cases resolved using "take ours" or "take theirs" merge strategies are the simplest. Next are cases that can be resolved by rearranging physical lines of code across the base, incoming, and current versions, followed by those that allow rearrangement of token sequences within a line. Finally, there are cases where developers introduce new tokens not present in the conflicted chunks. This classification respects the unique challenges of program merging, while you could also impose your syntax-based classifications on top.

"Propagation of the large-scale datasets for LM's training and testing." -> Can you correlate it to major LLMs training data cutoffs like what swe bench does? Once the benchmark is established major LLMs will surely respect it and not include it in the training split.

"DeepMerge and MergeBERT are essentially classification models that select answers from candidates, while our paper only focuses on LMs" --> This is not entirely accurate and needs to be corrected. DeepMerge composes resolution based on the line interleaving of the lines present in the incoming, current and baseline code chunks. While MergeBERT is a classifier, it first performs a token-level merge then selects from a set of composable patterns. I think the major point you are making about these two models is that they do not allow new tokens in the resolutions which are not present in the inputs.

"these methods do not consider the contextual content and are divorced from the actual scenario" -> MergeBERT does include surrounding code context lines, up to the max limit of the model's context window, similarly to your setting. These models are not open source afaik which is a good enough reason to not evaluate them in the paper. But the statement about them being "divorced from the actual scenario" is kind of overly bold and also incorrect.

## Minor comments
abstract:
"With the development of language models, researchers treat conflict code as text," -> Even before LLMs, actually, the ubiquitous git merge algorithm handles program merging as a line-level unstructured merge process, treating code purely as text agnostic to the language syntax or grammar. While AST-aware structured program merging algorithms have been developed, I see your point that unstructured text-based merging is more feasible path forward with the rise of LLMs.

introduction
"Git fails to resolve conflicts automatically" -> git does not attempt to merge conflicts automatically, if a 3-way merge attempt fails, the conflict markers would be generated, prompting user to resolve it manually

Main text:
"For LLMs, a simpler conflict does not necessarily mean it is easier to handle" -- what do you mean by "simpler" here?

---

### Official Review · Reviewer_5V4W · 2024-11-02

**Soundness:** 2
**Presentation:** 2
**Contribution:** 2
**Rating:** 3
**Confidence:** 4

**Summary:**

The work presented CONGRA, a dataset for evaluating automatic conflict resolution (ACR) tools. The authors aim to address limitations in existing benchmarks by introducing a complexity-graded dataset containing ~45k conflict cases from real-world projects. The built dataset includes conflicts categorized by type (text, syntax, and functional), which can be used for evaluating various conflict resolution models under differing complexities.  The authors evaluated several general-purpose and code-specific LLMs on this dataset and reported insights regarding the performance of specialized versus general LLMs.

**Strengths:**

+ A new dataset for evaluating ACR tools
+ Performance and limitations of existing LLMs on the new data were explored

**Weaknesses:**

- motivation of labeling conflict with the complexity scenario info is unclear
- dataset construction lacks transparency about the criteria used for project selection
- complexity type info (e.g., Text, Functional, Syntax) is too coarse-grained to be practical
- missing benchmarking existing machine learning-based ACR (e.g., DeepMerge, MergeGen)
- lack of deep analysis regarding the pros and cons of examined LLMs
- focused on limited syntax problems (i.e., declarations or definitions only)

**Questions:**

1. What are your criteria for selecting projects?

2. These machine learning-based ACRs also involved large-scale conflict datasets. Except for the complexity type information, are there any other differences between their dataset and the dataset created in this paper?

3. In 5.1, the authors explained why they are not compared to the existing baselines. They used RPredictor and MESTRE to illustrate the reasons for not comparing to machine learning-based ACRs. While some other machine learning-based ACRs, such as DeepMerge and MergeGen, are still comparable, as explained in 2.3, these ML-based ACRs are trained to offer resolution suggestions for merge conflicts. Why do authors exclude DeepMerge and MergeGen?

---

### Official Review · Reviewer_UqBD · 2024-11-02

**Soundness:** 2
**Presentation:** 3
**Contribution:** 3
**Rating:** 5
**Confidence:** 4

**Summary:**

The paper introduces **CONGRA**, a CONflict-GRAded benchmarking scheme designed to evaluate code merging capabilities of LLMs across varying conflict complexities. It proposes a novel method for classifying code conflicts into seven categories based on code operations, enhancing the understanding of conflict difficulty. Using this approach, the authors build a large-scale dataset of 44,948 conflict cases from 34 real-world projects written in C, C++, Java, and Python. They conduct the first comprehensive evaluation of LLMs on automatic conflict resolution using this dataset, revealing counterintuitive findings:
- LLMs with longer context lengths do not always perform better
- General purpose LLMs can outperform specialized code LLMs.

**Strengths:**

1. The construction of a substantial dataset containing 44,948 conflict cases from 34 real-world projects across multiple programming languages (C, C++, Java, and Python) is a valuable contribution to the field. Using real-world conflict cases from open-source projects is appropriate and lends credibility to the evaluation.
2. It introduces a new method for classifying code merge conflicts based on code operations extracted from syntax trees, enabling a complexity-graded dataset.
3. It provides insights into how different conflict types and contextual information impact the performance of LLMs.
4. The use of traditional metrics like accuracy and precision makes evaluation reliable instead of using automated frameworks like LLM-as-a-Judge.

**Weaknesses:**

1. The paper claims generalist LLMs outperform specialized code LLMs in automatic conflict resolution tasks. However, the code-focused models evaluated are relatively older and potentially less capable than the general-purpose LLMs used. Including recent code-specific LLMs would strengthen the authors' claims and provide a fairer basis for comparison.
2. The authors' explanations for their findings lack sufficient depth and appear speculative. For example, the claim that long-context models underperform because they are not adequately trained on merge conflict datasets is presented without substantial evidence or analysis (line 413). Incorporating more rigorous, possibly statistical, analyses would help substantiate these claims and make the conclusions more convincing.
3. For each task, different models are compared (eg. Table 2, Table 4, Table 6) which complicates direct comparisons of their performance. Populating these tables with the same set of models would help draw more insightful conclusions.
4. The authors use only the preceding and following code lines as context for resolving conflicts. Expanding the context to include cross-file information could potentially yield better results. Implementing methods such as tree-search combined with semantic similarity measures to identify and incorporate the most relevant context across files might improve the models' performance in conflict resolution tasks.
5. The authors propose a new dataset for evaluating automatic conflict resolution, however the dataset itself is not provided in the supplementary material.
---

If my concerns are addressed, I'm happy to revise my score.

**Questions:**

1. Given that the dataset is constructed from publicly available code repositories, how do you address potential bias in evaluation where LLMs may have encountered the "correct" resolution blocks during training?
2. As coding practices evolve, particularly with the growing influence of LLMs as coding assistants, how will CONGRA maintain its relevance as a benchmark since it is based on a static set of questions?
3. Sections 3.2 and 3.3 would benefit from clearer explanation of how the coloring process directly informs the classification methodology. Currently, these sections are not sufficiently clear for readers of all expertise. Could this workflow be elaborated?
4. What were the considerations in limiting the dataset to C/C++, Java, and Python? Is there potential for expansion to other languages such as JavaScript or Rust? Please consider adding this to Future Work.
5. Could the paper include visualizations depicting the distribution of conflicts across languages and conflict types to better illustrate the dataset composition?
6. The experimental setup using temperature=0.7 could be enhanced by either, a) Using near-deterministic settings (temperature~0) or b) Running multiple samples at higher temperatures. This would provide more statistically robust results.

---

### Official Review · Reviewer_qScG · 2024-11-03

**Soundness:** 3
**Presentation:** 3
**Contribution:** 2
**Rating:** 5
**Confidence:** 4

**Summary:**

This paper seeks to benchmark the applicability of large language models for the purpose of conflict resolution when merging git commits.
To that end, it proposes a pipeline consisting of:

1. The collection of merge commits with conflicts
2. 'coloring' of individual blocks containing conflicts into six categories (types, function body, function signature, variables, ...)
3. Classifying commits into 3 categories (text, functional, syntactic)
4. Applying six different large language models (each supporting context lenght of at least 8K) to let them merge the underlying conflicting commits
5. Metrics to assess the outcomes (normalized edit distance, winnowing, and cosine similarity)

This pipeline has been applied to 34 large open source projects, resulting in 23,334 conflict files and 44,948 'conflict scenarios' (which I assume are chunks in the code with a conflict).

Findings include that larger context support is not necessarily helpful, nor that code llms have advantages. Llama3-8B and DeepSeek-V2 appear to be the winners, with accuracy/precision hovering around 75% (depending on the programming language). Also, simpler conflict types are not necessarily easier to handle.

**Strengths:**

- Impressive merge conflict data collection established
- Paper establishes that LLMs with chain of thought can do a decent job in resolving merge conflicts

**Weaknesses:**

In section 3.1 I was confused about the way in which merges are detected.
It turns out this is explained on page 6 in the first paragraph of section 4.
I think this belongs in section 3, as it is independent of the nine systems chosen.

In Section 5 I was disappointed by the fact that comparison with existing merging tools was considered infeasible. While I can follow the reasoning to some extent, this does undermine the whole purpose of having a benchmark. In particular, the abstract states:

> we introduce CONGRA, a CONflict-GRAded benchmarking scheme designed to evaluate the performance of software merging tools under varying complexity conflict scenarios

I find it odd that existing software merging tools then are not taken into account.

The implicit hypothesis under this paper is that the content of the conflicting commits is enough to make merge decisions. However, this can also be a project-level decision, where e.g., feature A or feature B is considered more important.

Another assumption is that conflicts can be handled individually. But I'd say that choosing between A and B should be consistent -- one cannot pick some changes from A and others from B. The approach proposed doesn't seem to take this into account.

The findings suggest the 'context type' doesn't make much of a difference -- this can also mean that it is pointless to collect this type of information (as it doesn't truly affect the merge decisions).

The paper doesn't contain an analysis of whether the systems studied are part of the training data of the language models used. If so, the language models "know" what the eventual code should look like, which should help them.

Detail: Table 1 would be easier to follow if more meaningful names than Feature_1..6 were used

**Questions:**

- From how many actual (merge) commits are the 44,948 conflict cases?
- What are the pros/cons of handling conflicts individually vs considering all connected to a sigle merge attempt?
- What is the impact of training data on the results?
- What would it take to apply the dataset to the existing automated conflict resolution tools?

---

### Note · Authors · 2024-11-26

I have read and agree with the venue's withdrawal policy on behalf of myself and my co-authors.